# Humans Coexist, So Must Embodied Artificial Agents

## Abstract

This position paper introduces the concept of coexistence for embodied artificial agents and argues that it is a prerequisite for long-term, in-the-wild interaction with humans. Contemporary embodied artificial agents excel in static, predefined tasks but fall short in dynamic and long-term interactions with humans. On the other hand, humans can adapt and evolve continuously, exploiting the situated knowledge embedded in their environment and other agents, thus contributing to meaningful interactions. We take an interdisciplinary approach at different levels of organization, drawing from biology and design theory, to understand how human and non-human organisms foster entities that coexist within their specific environments. Finally, we propose key research directions for the artificial intelligence community to develop coexisting embodied agents, focusing on the principles, hardware and learning methods responsible for shaping them.

## 1 Introduction

Contemporary artificial intelligence (AI) systems have shown remarkable performance across diverse tasks such as the high-quality generation of data (image, text, video) [54, 2, 92], the creation of interactive world models [21, 4], and outperforming humans in complex decision-making tasks [127, 143, 140]. Fundamentally, three ingredients have been mostly responsible for this recent surge in performance: the creation of large-scale models [141, 32], the curation (or creation) of internet-scale datasets [123, 53] and a computationally-intensive offline training process [114, 157, 20]. This recipe has also been replicated for real-world robotic systems, resulting in the creation of large-scale datasets of expert-level interaction data in the real-world [106] and in simulation environments [145]. This approach has led to progresses in learning generalist robotic policies, able to perform a wide variety of manipulation and navigation tasks [14, 156].

As a community, we now envision concrete use cases of *embodied artificial agents*[1] for human interaction[2]. Despite their remarkable progress in controlled environments [18], embodied agents still struggle to gain a foothold in-the-wild scenarios [6]. Rodney Brooks' famous quip, "The world is its own best model" [110] is often used to encapsulate the problem of conceiving and deploying embodied artificial agents in the real-world [12]. However, we highlight that this challenge does not only emerge from the complex and dynamic nature of the real-world; it also stems from the constant tendency of viewing the real-world as an optimization problem [128]. Interaction *in-the-wild*, instead, is co-constructed with the humans *in-the-wild* [42], which is at odds with the dominant *problematize-solve-optimize-deploy* workflow of the contemporary AI community [59].

---

[1]We follow Paolo et al. [107] that defines embodied artificial agents as "agents that interact with their physical environment, emphasizing sensorimotor coupling and situated intelligence". Throughout this paper, we use the terms *agent*, *embodied agent*, and *embodied artificial agent* interchangeably for simplicity.

[2]In Appendix A we discuss the scope of interaction in the context of embodied agents.

We argue that the current approach to agent design is unsuitable for long-term, in-the-wild **interaction with humans.** In Section 2, we discuss why current embodied artificial agents are unable to cope with the strong dynamic nature of human interaction and the agents' inability to participate in its ongoing evolution. **We emphasize the need for a new paradigm for coexisting embodied agents**: mutable systems capable of continuously leveraging situated knowledge of both the user and the environment, highlighted in Figure 1, to establish meaningful and reciprocal interactions with the elements of its system. In Section 3, we formally define coexistence and its properties (situatedness and mutability) in the context of embodied artificial agents.

Our formal definition is complemented by a more practical approach for such agents to coexist in-the-wild. **In Section 4, we take an interdisciplinary perspective to understand how human and non-human organisms foster entities that coexist within their specific environments**. Therefore, we look to evolutionary biology and design theory, two fields that are epistemically grounded in the real-world [3]. There we highlight how biological organisms leverage properties of the real-world to take form during development (converge), and evolve in times of environmental changes (diverge). Similarly, we highlight how the double diamond design process (depicted in Figure 2a) has been an indispensable tool that has allowed designers and engineers to physically explore (diverge) and refine (converge) creative solutions. This section is not a call for the underlying systems of artificial agents to more closely resemble those of biological ones (for example, as in Darlow et al. [30]). Rather, to acknowledge the dynamic and interconnected nature of biological systems, as well as the existing methodologies in design theory that are adept with handling similar complexities. Taken together, they provide direction on building autonomous agents and our relationships with them.

**In Section 5, we highlight six key research directions for the AI community to develop coexisting agents**. We focus on the learning methods that enable coexistence, the hardware that sustains it, and the principles responsible for shaping it. Additionally, we discuss the ethical considerations in designing embodied agents that coexist with humans and play a role in shaping the future of human interactions. Finally, in Section 6 we contrast coexistence with alternative (predominant) viewpoints on embodied artificial agents. We see our work as a bridge, enabling the AI community to actively engage with the design research community in forging a path toward coexisting embodied agents.

## 2   Current Embodied Agents *Exist*

Recent advancements in perception, learning and hardware systems have enabled embodied agents to successfully perform complex actions in unstructured environments  [54, 2, 92].  We praise these advancements and believe that the current paradigm, based on multimodal foundation models for perception, reasoning and interaction, will be sufficient for these agents to *exist* with humans and within their environments.  However, we argue that the disregard of the issues pertaining to current embodied agents can have technical and cultural repercussions when employed widely in our societies. In particular, we focus on two fundamental properties of these agents: their *stagnant* nature, a consequence of having their abilities fixed at a specific moment in time, and their *generic* nature, due to their instantiation based solely on large amounts of pre-collected data. As current embodied agents are stagnant and generic, their widespread adoption risks conditioning the evolution of their interactions towards overly homogeneous ones, a phenomenon we denote by *steamrolling*.

### 2.1   Current Embodied Agents are Stagnant

Currently, we implicitly assume that there exists a predefined underlying data distribution, from which we can extract representative examples, to train and evaluate the behavior of embodied agents: for example, over the sentences people use when feeling happy, or over the possible socially accepted distances from humans while navigating a crowded room. Furthermore, it is assumed that this data distribution is static in time. As such, most of the knowledge acquisition and behavior exploration by the agent happens before it is deployed in a specific environment[4]. This inability to deal with changes in their own knowledge and their environment leads to their *stagnant* nature.

---

[3]Our position builds on past parallels between computers and biological processes [147, 19, 26].

[4]Recent approaches for training embodied agents additionally use fine-tuning to adapt the pretrained behavior of the agent to a specific task. We note that the fine-tuning data distribution is also itself predefined and static.

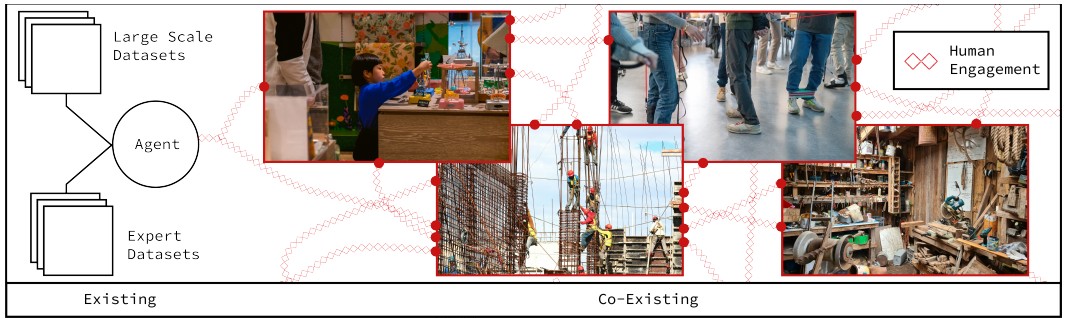

Figure 1: **Embodied artificial agents must coexist**. Current agents exist in the real-world, leveraging knowledge obtained from large-scale datasets and specific expert-level datasets to interact. We argue that embodied artificial agents must not only adapt to scenarios such as the ones pictured above but participate in their continual evolution. To do so, they must *coexist*, establishing meaningful and reciprocal relationships with the user and its particular environment by leveraging their diverse and situated knowledge.

Humans will adapt and change their behavior according to the environment they are situated in, but also participate in its shaping [33]. A classic example can be found in medical record cards in hospital beds: Nygren and Henriksson [103] found that the physical properties of the card (e.g., the handwriting, wear, tear, other marks) were contributing to the physician's decisions pertaining to both the patient and the activities surrounding their care. The hospital's culture and workflows are not converging to a "fixed" version, rather, they are perpetually evolving as the people, the environment and their interactions change. This is not only happening on a high functioning level: Vergunst and Ingold [142] show that even lower level motor skills, such as the way humans walk, are highly socialized and both culturally and contextually dependent. Therefore, a stagnant agent placed in this system would not be able to participate in this mutual shaping, as its behavior is a function of knowledge from a fixed point in time, which can be outdated at deployment time. Even a well-adapted agent at deployment will drift from the culture as the system evolves.

## 2.2 Current Embodied Agents are Generic

Recent advances in machine learning have enabled the extraction of general rules (e.g., grammar and social norms) from large-scale data to bootstrap the behavior of embodied agents [132, 154]. While learning general rules is valuable, we emphasize the crucial distinction between being generic and being general. *General* knowledge captures fundamental principles that apply broadly across a wide range of cases, enabling generalization, a desirable feature in both autonomous systems and humans [5]. In contrast, *generic* knowledge is applied across many situations without accounting for their specific nuances or contextual diversity. By learning generic information from large-scale datasets, agents reinforce (potentially harmful) biases that exist on such data [109]: for example, image generation models produce images of white men for the prompt "a software engineer" and women with darker skin tone for the prompt "a housekeeper" [13]. Current embodied agents, which often employ such models for interaction purposes, also rely on generic knowledge[7]. Exploiting only generic knowledge is also inefficient. For example, compare a highly controlled space such as a factory, where workbenches and machines are specifically configured, to a home or office space. Each instance of a home or office is unique and contains situated knowledge that is specific to its configuration and the humans in it [33]. An agent that relies solely on generic knowledge, is at a clear disadvantage against an agent that also exploits situated knowledge and, just as importantly, contributes to the ongoing exploration and exploitation of culture and workflow in the space [46].

## 2.3 Current Embodied Agents Will Steamroll

When a stagnant and generic agent is placed in a dynamic environment, either the agent will become obsolete and removed from the system over time [6], or the humans will adapt to it. Therefore, in the latter case, human cultures and workflows will start to converge towards those dictated by the agent,

---

[5]In Appendix A we discuss the challenges of generalization for current embodied agents.

[6]See Appendix D for the case of the *novelty effect* in the current deployment of social robots.

limiting the exploration and discovery of novelty [135]. We denote this phenomenon as *steamrolling*. Whilst the current prevalence of artificial embodied agents in everyday settings limits the empirical evidence to support steamrolling, we point to convergent effects in similar socio-technical spaces that foreshadow steamrolling. Geng and Trotta [44] estimates that 35% of all scientific paper abstracts in computer science are now written in "LLM-style". Székely et al. [133] argue that such a style is spreading from text interactions to more embodied interactions such as speech. Meincke et al. [94] have shown that in brainstorming sessions, despite having a positive impact on individual creativity, the pool of LLM-assisted responses exhibits lower diversity. They argue that effective brainstorming is undermined by the collective use of LLM-tools because instead of creating diverse ideas, similar thoughts were repeated by multiple participants. In the context of embodied artificial agents, we expect steamrolling to inhibit divergent behavior, in favor of reinforcing already existing behavior of both human and agents. We argue that steamrolling will also affect the future capabilities of the agents we develop: a model trained on a progressively narrower distribution (such as data curated from its own outputs) suffers from rapid degradation in the quality of its generated output [126].

## 3   Future Embodied Agents Must *Coexist*

Long-term interactions between humans and embodied artificial agents have been extensively studied by the robotics community [85, 31, 79], focusing on specific properties of the interaction such as acceptance [31], engagement [115, 84] and disclosure [98, 89]. Here we take a holistic view of the long-term interactions of embodied agents within a system and provide a general-purpose, formal definition of coexistence[7].

> **Definition**: An embodied artificial agent is *coexisting* in a system if it sustains meaningful and reciprocal interactions with humans and their environment over time.

Consider a system $S = \{A, H, E\}$ consisting of an embodied agent $A_t$ present in a specific environment $E_t$ alongside a human user $H_t$, at a given time $t$. There exists a quality function $Q_O(t)$ that overall describes the system and its evolution, measured from the point of view of an observer $O \in S$. The quality function is influenced by the interactions between the agent, the user and the environment[8]. We note that the goal of the agent does not necessarily align with this quality function as it may be independent of its intended task (e.g., a household robot assisting with chores may perform its tasks efficiently but disrupt the human's workflow and create frustration).

We can define two categories of interactions within this system. A unilateral interaction $X_t \to Y_t$ occurs if the state of element $Y$ of the system at the next time step $(t + 1)$ is influenced by element $X$, while the next state of $X$ remains independent of $Y$,

$$Y_{t+1} = f_Y(Y_t, X_t, y_t, x_t), \quad X_{t+1} = f_X(X_t, x_t), \tag{1}$$

where $f_X, f_Y$ are unknown and dynamic transition functions, and $x_t, y_t$ are the actions of $X$ and $Y$ at time $t$. Similarly a *reciprocal* interaction $X_t \leftrightarrow Y_t$ occurs if the next state of both elements are mutually influenced,

$$Y_{t+1} = f_Y(Y_t, X_t, y_t, x_t), \quad X_{t+1} = f_X(X_t, Y_t, x_t, y_t). \tag{2}$$

Interactions influence the long-term quality of the system, which can be measured after a (system-dependent) time horizon threshold $T_S$. We define a *meaningful* interaction as one that, given sufficient time (i.e., in the long-term), does not decrease the overall quality of the system, as evaluated by all elements of the interaction, compared to the absence of such interaction. Formally,

$$\exists T_S > t, \forall t' > T_S, \forall O \in \{X, Y\} : Q_O(t' \mid X_t \to Y_t) \geq Q_O(t' \mid \emptyset), \tag{3}$$

where $\emptyset$ denotes no interaction and the conditional quality function $Q_O(t' \mid X_t)$ indicates the value of the quality function at $t'$ given that process $X$ occurred at $t < t'$. A coexisting agent $A^*$ is then defined as an agent able to maintain reciprocal and meaningful interactions in the long-term. Intuitively, this means that, in the long run, the agent benefits the system more than its removal would,

$$\exists T_S > t, \forall t' > T_S, \forall O \in \{A^*, H\} : \tag{4}$$

$$Q_O\big(t' \mid A_t^* \leftrightarrow (H_t, E_t), H_t \leftrightarrow E_t\big) \geq Q_O\big(t' \mid H_t \leftrightarrow E_t\big).$$

---

[7]In Appendix B we present additional considerations and limitations of our definition of coexistence.

[8]In Appendix C we discuss in depth how to potentially instantiate and measure the quality of the system.

## 3.1 Properties of Coexisting Embodied Agents

**Situatedness**  A coexisting agent $A^*$ should actively leverage the fact that it is situated within a specific environment and exploit the unique situated knowledge embedded in the user and their environment, rather than relying solely on pretrained knowledge. This capability reflects the agent's *speciation* to its particular system. Formally, this can be expressed as:

$$\exists T_S > t, \forall t' > T_S, \forall O \in \{A^*, H\}, \forall O' \in \{A^*, H'\} : \tag{5}$$
$$Q_O\big(t' \mid A_t^* \leftrightarrow (H_t, E_t)\big) \geq Q'_{O'}\big(t' \mid A_t^* \leftrightarrow (H'_t, E'_t)\big),$$

where we define a distinct system $S' = \{A^*, E', H'\}$ with its own specific quality function $Q'_{O'}(t)$, but involving the same agent. Note that, contrary to the generic nature of current embodied agents, we argue that the behavior of coexisting agents should improve the quality of their specific system, even if the same behavior would result in a overall quality decrease in other distinct systems.

**Mutability**  A coexisting agent $A^*$ should be capable of continuously adapting its behavior while also influencing the behavior of other elements within the system. Formally, this adaptability relates to the concept of reciprocal interactions:

$$\exists T_S > t, \forall t' > T_S, \forall O \in \{A^*, H\} : \tag{6}$$
$$Q_O\big(t' \mid (H_t, E_t) \leftrightarrow A_t^*\big) > Q_O\big(t' \mid (H'_t, E'_t) \to A_t^*\big).$$

This condition implies that coexisting agents and humans should be able to mutually shape each other in ways that enhance the overall quality of the system. In contrast, the stagnant nature of current embodied systems often requires a unidirectional training of the human user.

Importantly, changes in the agent's behavior do not always lead to an immediate improvement in system quality and may sometimes have the opposite effect. As discussed in Section 4, coexisting agents must be capable of generating divergent behavior even within a closed system. This ability is crucial for the long-term success of the system as it enables the exploration of alternative solutions, not only in the agent's behavior but also in how its behavior impacts the other elements of the system.

# 4  Coexistence Elsewhere

Beyond the formalism of coexistence, the question remains of how to instantiate embodied agents with the ability to coexist in the real-world[9]. To address this challenge, we take an interdisciplinary approach to understand how humans and non-human organisms foster entities that coexist at different levels of organization: from the processes of biology to the methods of design theory. We explore research in these fields that highlight the value of mutability and situatedness in fostering coexistence.

## 4.1  Coexistence in Biology

Biological systems offer a unique perspective on coexistence, showing how living organisms evolve, adapt, and sustain themselves in their own environments. Unlike current embodied agents, which assume that all necessary knowledge can be extracted from data and encoded, biology balances encoded information with meaningful interactions with the physical world to shape adaptation, survival and purpose. In this section, we present examples from genetics and developmental biology that explore how biology navigates this balance.

**Not everything is in the genome**  Underlying the majority of machine learning models is the assumption that all necessary knowledge to act/decide optimally can be extracted from data and subsequently exploited. However, biology provides a perspective shift in regards to the nature and role of data in the evolution of agents. To illustrate how encoded information is only one part of what shapes biological organisms, we turn to the Human Genome Project [27]. When this project successfully sequenced the entire human genome it was widely believed that the genome could define what humans are, an "instruction book for life". However as Ball [9] explains, the project instead marked the beginning of a paradigm shift in biology that de-throned the genome as an encrypted

---

[9]In Appendix D, we examine if current agents already coexist, providing examples on why they fall short.

source of life's secrets. Instead it was shown that an organism is not only defined by the genome but also by principles of self-organization that are enacted by being situated in the physical world [9].

A striking example of this new reality can be seen in developmental biology, where the number, thickness, and size of a rodent's digits were not found to be encoded in the genome. Instead, a timing of particular proteins (namely BMP, SOX9 and WNT) that disperse in physical space determines the number of digits and the space between them. Raspopovic et al. [116] discovered that they could manipulate the activity of these proteins and could thus influence the number of digits formed and their thickness. This example shows how the characteristics of the physical world play a role in defining information and intelligence, providing an extremely efficient way of acting in the world [9].

**Biology is not an optimizer**   Leveraging the physical world is not only about converging on optimally efficient solutions but also about diverging from locally competitive landscapes. It is a common misconception that biology is an optimizer. As Stanley and Lehman [128] write: "Early evolutionists believed, and indeed many non-experts still believe, that evolution is progressive, moving towards some sort of objective perfection, a kind of search for the über organism". In fact, "most evolutionary changes at the molecular level [DNA] are caused not by Darwinian selection but by random *genetic drift* of mutated genes that are selectively neutral" [149].

As an example, consider the protein HSP90, where HSP denotes for "heat shock protein". HSP90 was discovered to have a kind of plasticity modulation effect on the body plans of the common fruit fly. In warmer conditions, this protein enables more variation in the morphology of the fruit flies, in places such as its abdomen, bristles, eyes, legs, thorax and wings [120]. In addition, these traits were able to be passed down immediately to the next generation [149]. It is argued that processes like the ones observed here played a large part in periods of intense diversification in living organisms during the Cambrian explosion [9]. This alludes to the idea that evolution, whilst highly divergent, is both *bound* and *liberated* by the laws of nature: by using existing building blocks in creative ways, it is able to keep a tension between convergence and divergence [45], conditioning and stimulating exploration and exploitation of novel solutions within its own laws.

## 4.2   Coexistence in Design

We have seen how biological organisms exploit being situated in the world to balance convergence and divergence in order to foster coexistence in their physical setting. However, how a human could instantiate a similar process, with their plans, goals, morals, and aesthetics is still unclear. The answer lies in the divergent and convergent *processes* of design which cause an individual to engage reciprocally with technology and its environment, as highlighted in Figure 2.

**The double diamond**   The design process often converges to a design outcome, due to performance specifications [29], or intended functions or styles [119]. In order to deliver an outcome, methods and heuristics exist within each design discipline [136, 29]. But beneath these formalizations lies a practice that is tacit and with an improvisational dimension. This dimension is not only a function of expert knowledge from formal education (industrial, mechanical, electrical, graphical, architectural, etc.), but a craft-like knowledge of their materials, and a situated understanding of how to use them, built up over years of experience [124]. This process is popularly characterized by the UK Design Council's double diamond [125], highlighted in Figure 2a. Initially when a designer receives a specification, they begin to explore *divergently* how to think about the problem: this involves reasoning about the materials, context, people, social structures, and policy context of the request [136]. Subsequently, they begin to *converge* on a more concrete definition of the problem and present it to the stakeholders involved. At this moment, all stakeholders *diverge* again, exploring various designs without limits as they explore the potential solution space. Finally, the designer converges on a solution, synthesizing all that they have learned to present a design that is on time, budget, and to specification. The double diamond merges a designer's expertise with their situated knowledge and experience.

The outcome-centered perspective inherent in the double diamond brings with it the notion that a design should be finished and then deployed in its "finished state" [137, 117]. Here we find an interesting bridge to current embodied artificial agents: they too pass through a phase of training and are only subsequently deployed when they have reached a pre-defined threshold of performance. In interaction design, this perspective limits a finished design to its intended function. Despite the efforts of human factors, user-centered design and participatory design methods [125], ethnographic

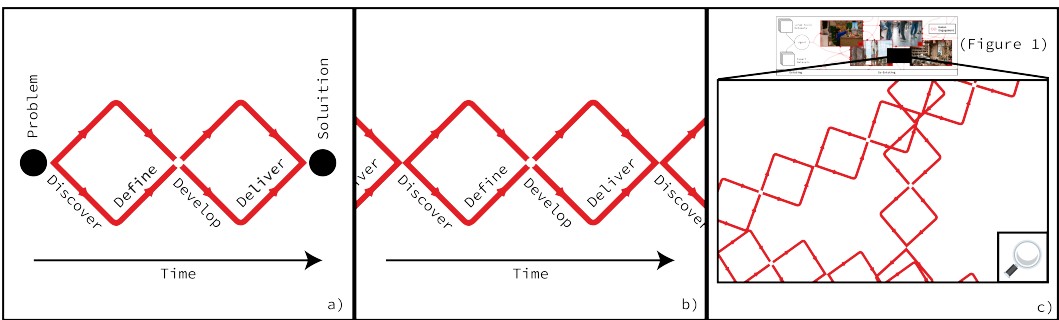

Figure 2: **The evolution of coexisting embodied agents**: a) The double diamond process, with its distinct problem/solution-focused beginning and end; b) Removing the head and tail off the double diamond reveals a continuous and reflective engagement with technology as demonstrated by the field of research through design; c) Revisiting Figure 1, by involving end-users as design researchers, they are encouraged to draw from their experience to integrate technology into existing contexts and to actively shape and explore new ones.

studies often reveal the user to be constantly spending time and creative energy to configure these finished designs and their intended functions into their own lives [33, 129, 36, 102]. This has lead to the increasingly blurred line between what constitutes a designer and a user of technology [117].

**Research through design (the continuous double diamond)**    The field of human-computer interaction (HCI) has seen in the last two decades the rise of *research through design* (RtD) [69, 43] which supports the notion that design is never finished. It is commonly framed as "an active process of ideating, iterating, and critiquing potential solutions, design researchers continually reframe the problem as they attempt to make the right thing" [160]. RtD can be understood as a continuous double diamond (see Figure 2b), with its tail (problem) and head (solution) lopped off. The design process then becomes reflective: where the morals, lived experience, and aesthetic preferences of the designer[10] can inform their professional training [76], leading to completely new (divergent) ways of interacting with technology [11], or familiar (convergent) twists on existing ones [104]. This kind of continuous design has been termed "drifting" by Krogh et al. [72] and bears a striking, functional resemblance the genetic drift discussed in Section 4.1.

### 4.3    From Elsewhere to Embodied Agents

By exploring coexistence in biology, we have shown that living organisms leverage the physical world to offload the need for encoding all necessary information for survival and action, while also enabling diverse and adaptable behaviors. By exploring coexistence in design, we have highlighted RtD as a promising approach to balance convergence and divergence in the interaction between humans and technology. A common thread between these explorations is the notion of drifting, a process that allows systems at different levels of organization (from microorganisms to human designers) to cope with changing objectives (mutability) across different environments (situatedness) over time. In the next section we extend these ideas to embodied agents: leveraging the situated knowledge in the environment and in the human user enables embodied agents to successfully change, evolve and interact in a meaningful way within their specific environments.

## 5    Towards Coexisting Embodied Agents

We have seen how both human and non-human organisms evolve and coexist within their own environments. What can the AI community learn from these processes? **This section outlines six key research directions toward fostering coexistence and developing coexisting agents**, focusing on the *principles* that shape coexistence (1, 2), the *hardware* that supports it (3, 4) and the *methods* that enable it (5, 6). Finally, we address ethical considerations of coexistence.

**1) Foster coexistence by embracing *open-endedness***    Hughes et al. [58] argues for open-endedness to design continuously evolving agents, defining it as a property of systems that produce novel

---

[10]As a "first person method" [91] RtD can trace its theoretical foundations to the theories of embodiment [81].

and learnable artifacts from the perspective of an observer. We propose that open-endedness is an essential principle for the design of coexisting agents. They should be able to continuously evolve with their environment, changing with it and contributing to its change. This means that both the agent as well as its objectives are non-static, creating the need for open-endedness. We agree that open-endedness is essential to achieve coexisting agents, and highlight the shared importance of the observer's perspective between open-endedness and RtD. We see the role of the observer as a driver of continuous change and exploration, not just a creative optimizer for a given task.

**2) Foster coexistence by embracing the *user as the designer***   Often the user is seen as someone who should not have to deal with the complexities that arise from interacting with technology [102]. In RtD, this perspective is rejected in favor of seeing the user as someone who has situated knowledge, or is a *connoisseur* of their situation [160, 91]. Situated knowledge includes tacit, institutional, craft or social knowledge, and can help mediate an agent's purpose or behavior in an environment. We argue that this perspective is essential to coexistence and should guide the development of embodied agents. In Appendix F we provide some examples that demonstrate the potential of this principle.

**3) Foster coexistence in the space *around* the agent**   Consider an agent using an inside-out navigation system, (e.g., SLAM [35]) which is inherently prone to drift. If an outside-in navigation system is instead used (where the agent navigates relative to a set of beacons), the agent can be designed such that the situated human can configure the placement of the beacons. Whilst this sounds like a poorly designed system that requires constant maintenance[11] research on AI education has favored this more active and experiential approach, as it fosters a kind of tacit understanding of the capabilities and limitations of the system [41, 150, 63]. The situated knowledge gained from this approach can help users coexist with agents in a specific environment.

**4) Foster coexistence within the *morphology* of the agent**   Evolutionary robotics has demonstrated that by changing the morphology of an artificial agent, you change their capabilities and limitations [110]. Additionally, advancements in manufacturing technology are rapidly expanding the potential forms an agent could take [70]. This concept has been explored in the context of human-drone interaction. La Delfa et al. [77] gave users a drone that could initially only hover in place. By moving with the drone, the users were able to selectively expand its perceptive field. As the field grew in size, unique patterns of interaction emerged based on its the shape and size. The mutability of the drone's sensory field allowed for a meaningful relationship to evolve.

**5) Foster coexistence by using foundation models as *external* components**   Recent methods have used foundation models or composite systems that incorporate foundation models to generate agent behavior [18]. While using these models directly as policies is not sufficient for coexisting agents, foundation models still have valuable properties that can be leveraged (even if these are currently prone to hallucinations [88, 159]): they can act as an external storage of *generic* knowledge that an agent could query for bootstrapping purposes *without replacing situated knowledge*. This external knowledge base could help decrease the memory and computation requirements to build embodied agents [40]. Additionally, foundation models could serve as external teachers to agents to bootstrap their performance [151] and guide exploration [74] *without replacing situated exploration*. While we understand these models can also be used for multimodal perception and reasoning, we highlight the risk of embedding such internal components of embodied agents with generic and stagnant knowledge and encourage researchers to consider using the real-world as "its own best model"[110].

**6) Foster coexistence by learning and evolving with humans *as we go***   In their current form, even common learning approaches designed to overcome the assumption of a static optimization problem (e.g., online reinforcement learning, meta-learning, and continual learning) are insufficient to foster coexistence[12]. To enable mutability and speciation, instead, we advocate for human-in-the-loop learning with evolutionary algorithms [86, 22]. Evolutionary algorithms [8, 87] can maintain diverse candidate solutions throughout the (continuous) learning process, allowing agents to execute multimodal behavior, both divergent and convergent [97]. When combined with interactive learning paradigms [155, 96], such as by using preferences or demonstrations, these evolutionary processes

---

[11]As a comparison, we would like to highlight the resources required to create and curate large-scale datasets.

[12]For an extended argument on why this is the case we refer the reader to Appendix E

can also be progressively shaped through meaningful interactions with the human, allowing the agent to deal with evolving goals and expectations.

## 5.1 Should We Foster Coexistence?

Coexistence gives users the ability to shape and be shaped by embodied agents, carrying the inherent risk of manipulation of the agent's behavior by malicious users [100] and vice versa. However, when users are given the responsibility to shape the agents in their environment, we enable them to do so in their own particular way, resulting in a *heterogeneous* population of bespoke agents. In contrast, Székely et al. [133] and Bongard [15] warn of the risks of malicious manipulation of human users using homogeneous artificial intelligence agents at internet-scale. We still highlight the importance of developing agents that have the ability to recognize harmful behavior and respond in a manner that upholds safety, fairness, and accountability.

The heterogeneity inherent to coexistence, requires a continuous effort and can be a slower process compared to unidirectional alignment. It also involves high level of ambiguity (e.g., the exploration of a solution space without a set objective) and an active, reflective engagement on behalf of the end-user (e.g., the evaluation of said exploration). Similarly, local and diverse AI collectives have been shown to exhibit strong innovative capabilities and pro-social behavior [80]. Still, they should not be treated as the same kind of relationships: we highlight the need for ethical and legal frameworks that elevate human well-being and safety above that of artificial agents.

## 6 Alternative Views to Coexistence

**AGI/ASI vs. coexistence** While coexistence is a goal and property in itself, other positions argue for different goals and capabilities of long-term interactive artificial agents within our societies. Paolo et al. [108] argue in favor of attempting to achieve artificial general intelligence (AGI), describing the goal as "creat[ing] intelligence that either parallels or exceeds human abilities". They state that embodiment and situated intelligence are essential conditions for achieving AGI. Similarly, Hughes et al. [58], argue in favor of artificial superhuman intelligence (ASI) and propose open-endedness as a prerequisite to ASI. Whilst we share an understanding of the importance of embodiment and open-endedness, neither position requires mutual co-shaping for the widespread use of artificial agents in human society. Despite its risks [99, 93], AGI and ASI proponents point to the accelerated progress and benefit for humanity driven by a single superior intelligence. Instead, we believe that through the increase in diversity, coexistence aims for something more beneficial and robust: we place meaningful and reciprocal interactions with humans at the center of our proposal.

**Unilateral alignment vs. coexistence** Yang et al. [152] state that "unified alignment between agents, humans and their environment" is key to the success of agents in real-world applications. They propose that agents not only align with human users, but also with the environment and the agent's own constraints. Furthermore, they highlight the difficulty of discovering human intentions due to partial observability, temporality and stochasticity. Although they discuss the need for agents that can align with evolving preferences, a process they denote as continual alignment, they still assume that preferences are something that is known by the human a priori. They write: "the tasks assigned by humans can be viewed as the initial inputs to the working system (especially to the agents), which reflects the underlying goals and human intentions". We instead believe that the human's goals are formed through interacting with the agent.

## 7 Conclusion

In this paper, we have argued that the current paradigm for designing embodied artificial agents is fundamentally ill-suited for long-term, in-the-wild human interaction. We proposed *coexistence* as a new paradigm for the design of embodied agents that emphasizes meaningful, reciprocal interactions sustained over time. Drawing from biology and design, we showed how human and non-human organisms leverage the physical world in convergent and divergent ways. We outlined key research directions for coexisting agents, emphasizing open-ended, human-in-the-loop learning and the user's role in shaping both behavior and morphology. We envision a future where artificial agents do not just exist but *coexist*, actively shaping and adapting with humans and their environments.

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

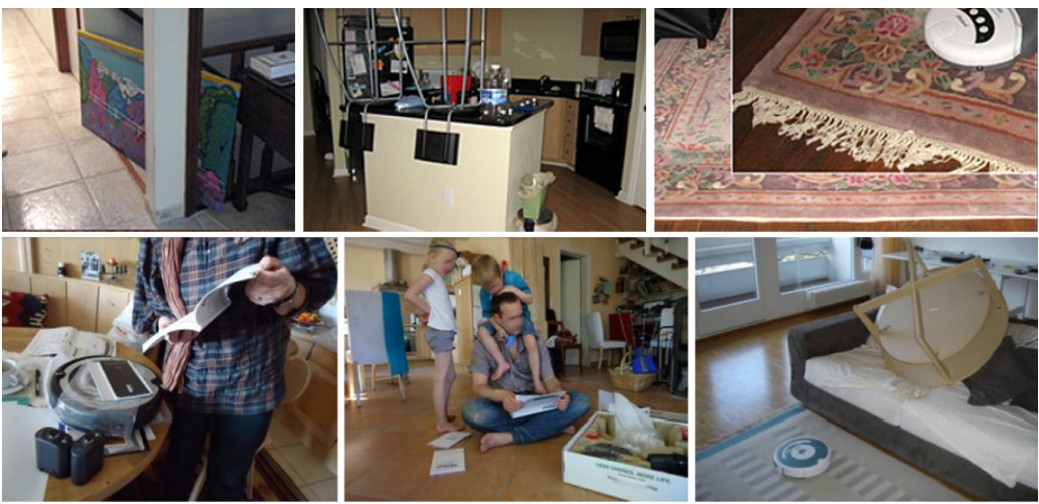

Figure 3: Fink et al. [38] and Sung et al. [130] highlight the various ways users modify their behaviors and living spaces to integrate autonomous robot vacuum cleaners into daily life.

## A    Additional Notes on Coexistence

**Interaction with humans in coexistence**    When discussing coexistence, one might ask about embodied agents that are not directly interacting with humans. Does coexistence apply to them as well? Why would an embodied agent that has no relationship with humans have to be designed with interaction in mind? We take the perspective that embodied agents that were made by humans are *interacting* with humans, even in cases where such interaction is not evident at first glance. Supporting our perspective is Frauenberger's [42] seminal review article on the shifting theoretical foundations of the field of human-computer interaction. Frauenberger [42] states that there are no clear boundaries that separate humans from technology. Instead, the purpose of a technology (e.g., an embodied agent such as a robot) is constituted by not only its physical self but its environment and the surrounding human and non-human agents. Therefore, the vast network of interconnected technologies that support our every day lives are said to be *entangled*, and expands the notion of interaction beyond dyadic "turn taking".

Take for example, an autonomous robot vacuum cleaner: from an earlier interaction perspective, its sole purpose is to clean up after humans. From an entanglement perspective, however, a robot vacuum cleaner is not seen as a simple tool for cleaning, but has an effect that goes beyond that of its original purpose. This is demonstrated by Sung et al.'s [130] field work with robot vacuum cleaners. Their work frequently revealed users cooperating with the robot by preparing the space before it was cleaned, modifying their environment to facilitate easy navigation and even socially interacting with it, as highlighted in Figure 3. Therefore, despite most of the robot's actions taking place in the absence of any human, the effect of these actions impacts other objects in the home (e.g., moving furniture, routing cables), human practices (e.g., the habit of tidying before the robot begins its operation) and the environment itself (e.g., the installation of threshold ramps in doorways). However, we note that the effects visible in Sung et al. [130] do not, by themselves, constitute coexistence, but only entanglement. For example, the subsequent in-the-wild study on robot vacuum cleaners by Fink et al. [38] draws attention to several cases of users unable to adapt to the robot, as their existing environment and practices were not amenable to the inflexible design of the robot. This highlights how, by designing robot vacuum cleaners with a fixed and narrow purpose (i.e., cleaning after the human), we are inhibiting the continuous evolution of its role and effects, and, therefore, coexistence. We refer the reader to Appendix D, Social robots, for a complementary discussion that focuses more on the social aspects of everyday life with robots.

**Isn't coexistence just optimization?**    With coexistence, we are proposing a form of learning and development that is not as strictly focused on optimization as current approaches. In classic machine learning, a task is defined by an objective function, which captures what the model/agent should do, and then fitting available data to that objective. The agent learns by optimizing towards this objective. In our position, we provide a different perspective on this practice. We broaden the focus

by paying attention to system-wide quality (motivated by contemporary views on what interaction is, see previous paragraph). In order to uphold this quality, it might occasionally be necessary to go against the objective function, or to acknowledge the fact that a *static objective function for long-term interaction may not even exist*. While classic machine learning approaches try to converge to a solution, we emphasize the necessity to convergence *and diverge*. Optimization has a place in coexistence, but it needs to be balanced with continuous exploration. Without it, the ability for humans and artificial agents to conceive novel objectives (read, to change and evolve as a society) will be severely distorted (see Section 2).

**Designing frameworks for coexistence**  Considering the design of frameworks that implement coexistence in practice, we aim to embrace a continuous process that balances divergence and convergence, inspired by biology and design. As we outline in Section 4.2, the process that produces the design itself (the continuous double diamond) will result in coexisting entities. Given the improvisational and situated nature of RtD, any kind of framework that implies a strict procedure would go against the value of coexistence. Guidance on these kinds of processes is given by Pfeifer et al. [111], who define design principles for Intelligent Systems, as well as Krogh et al. [73], who identify and analyze drifting as a property of continuous design processes. La Delfa [75, p.111] outlines how these process can be combined in the context of embodied artificial agents.

**The role of generalization in coexistence**  As discussed in Section 2.2, generalization – the ability to leverage *general* knowledge to act across similar scenarios – is a desirable feature for embodied agents. However, we would like to highlight the challenges and potential pitfalls of achieving generalization at the cost of employing *generic* knowledge. Consider once again, the example presented in Section 2.1: Nygren and Henriksson [103] found that the physical properties of medical cards of the patients (e.g., the handwriting, wear, tear, other marks) were contributing to the decision-making of the physician. Achieving such nuanced care and attention from embodied artificial agents through generalization would be unfeasible, as it would require the collection, and inclusion in the training procedure, of increasingly fine-grained physical and social cue data. A similar observation is made by [139], where the authors reveal that zero-shot generalization of concepts in multimodal models require an exponential increase in progressively more fine-grained data. One could argue, then, that we should instead focus on developing increasingly realistic simulators. However, such option also appears fundamentally unfeasible. Bharadhwaj [12] argues that "even the best simulators cannot match reality", concluding that "scaling simulation frameworks is unlikely to directly help with these [manipulation] tasks as each of these would require separate nuanced considerations for faithful simulation", hindering generalization. We highlight that the impossibility of collecting exponentially fine-grained data in the real-world to pretrain large-scale models, alongside the infeasibility of developing faithful simulators of the real-world, further motivates our argument to, instead, develop agents that are able to continuously leverage the physical and social cues *present in their specific environment* to achieve coexistence.

# B  Additional Notes on the Definition of Coexistence

**Nature of $Q_S$ and $T_S$**  Like all the elements in the system, the operationalization and interpretation of the quality function $Q_S$ is dynamic (meaning it changes over time) and specific to every system. In Appendix C we discuss some potential proxy metrics for the quality function. The same can be said of the time horizon $T_S$: each particular system should have, even if implicitly, a specific time horizon to assess the evolution of the system itself.

**Assumptions of coexistence**  For simplification we have implicitly assumed that our system is *closed*, meaning that the quality of the interaction is only influenced by the elements within the system (environment, human and agent). We have also assumed that there is a single human user in the system. However, we can easily extend this to open systems and multiple users by considering the correspondent interaction terms with additional elements external to the system (e.g., external societal rules, other human and agent members of a team), without a significant change on the definition of coexistence.

Formally, consider a system $S = \{A^*, \mathbf{H}, \mathbf{A}, E\}$ where $\mathbf{H}_t$ represents the state at time $t$ of the set of all the humans in the system and $\mathbf{A}_t$ the state of all additional agents in the system. Let $\mathbf{E}_t$ be

the state of the set of external components to the system (e.g., external societal rules or observers imposed on the system). We still formally define a *coexisting* agent $A^*$ as,

$$\exists T_S > t, \forall t' > T_S, \forall O \in \{A^*, \mathbf{H}, \mathbf{A}\} :$$
$$Q_O\left(t' \mid A_t^* \leftrightarrow (\mathbf{H}_t, \mathbf{A}_t, E_t),\ \mathbf{H}_t \leftrightarrow E_t,\ \mathbf{H}_t \leftrightarrow \mathbf{A}_t,\ \mathbf{A}_t \leftrightarrow E_t\right) \geq \qquad (7)$$
$$Q_O\left(t' \mid \mathbf{H}_t \leftrightarrow E_t,\ \mathbf{H}_t \leftrightarrow \mathbf{A}_t,\ \mathbf{A}_t \leftrightarrow E_t\right).$$

We would like to point out that, despite the fact that the system is open, and influenced by the interactions with $\mathbf{E}$, these external components of the system are not observers of the quality function $Q_O$ of the system. In practice, this means that their assessment of the overall quality of the system does not influence the quality of the system itself, as it is measured only by the elements that belong to the system. As an example, consider a companion robot interacting with a teenager ($\mathbf{H}$). Parents ($\mathbf{E}$) might disapprove of the informal language style used by the robot. However, their external assessment does not directly affect the quality of the system: instead, it is measured exclusively based on the perceptions and experiences of the teenager and the robot. In another instantiation of the system, one that encompasses both the teenager, the robot and the parents, the latter would naturally influence the quality of that system.

**The quality of a system is not monotonically increasing**   We do not expect the quality of the system to be monotonically increasing over time; in fact, *we argue that it should not*. Formally, there may exist time steps $t_1 < t_2$ such that $Q_O(t_2) < Q_O(t_1)$, even in the case of meaningful and reciprocal interactions. The requirement for coexistence concerns the asymptotic behavior of the quality function over a system-dependent horizon $t' > T_S$.

**Credit assignment in coexistence**   A core challenge in evaluating coexistence lies in the credit assignment problem [112], i.e., determining which interactions are responsible for the changes in the overall quality of the system over time. This challenge is exacerbated by the fact that the effects of an interaction may not immediately be observable, as multiple overlapping interactions may contribute to a shared outcome. Adding to these challenges, the quality function itself is not stationary, due to the evolution of the human, agent and their environment.

Assigning credit for the long-term impact of meaningful and reciprocal interactions remains an open research problem. Future work may explore the use of causal inference [95] or multi-agent reinforcement learning techniques [146] to better disentangle the contributions of individual interactions.

## C   Measuring Coexistence

One fundamental challenge of implementing coexistence for embodied agents is how to measure the quality function $Q_S$ of a system. As briefly mentioned in Section 3, the human-robot/computer interaction community has proposed several (often complementary) self-reported metrics to evaluate the quality of an interaction. For example,

- **Trust** reflects the human belief that the robot will behave reliably, safely, and as expected [65]. In collaborative tasks, the performance and attributes of the agent strongly influence the user's trust. Trust, in turn, correlates with better team performance and interaction outcomes [51].

- **Engagement** reflects how involved, attentive and interested the human is during the interaction. It captures the degree of active participation in the interaction by the human user (e.g., paying attention to the agent, responding to it, initiating the interaction) [105].

- **Likeability** measures how pleasant, friendly, and likeable the agent is to the human user during the interaction. Likeability can emerge from the personality of the agent, its appearance or its behavior [121, 134, 28].

- **Social Presence** relates to the feeling that the agent is a social entity present in the system, as opposed to being a passive machine or tool [24]. A strong sense of social presence usually indicates a more natural and engaging interaction, which can in turn increase trust and empathy towards the agent. Conversely, if the agent is seen as having zero social presence, the user might not engage socially or might not heed social cues from the agent.

- **Acceptance** measures how willing users are to interact with the agent and integrate it into their routines [37]. As such, it is an important proxy for the overall quality of the system, as it gauges the possibility of maintaining long-term interactions within the system.

One fundamental issue with the measure of these self-reported metrics is the fact that they are traditionally uni-directional, i.e., from the perspective of the human. They are usually collected via questionnaires or interviews before and/or after an interaction, capturing the human's personal experience, attitudes, and perceptions of the agent or the interaction. Moreover they are intrinsically task-dependent, as they either measure the quality of a specific interaction, or use the result of a given interaction to condition the human to predict the quality of future interactions.

Besides self-reported metrics, researchers in human-robot and human-computer interaction often also rely on objective behavioral interaction metrics, i.e., what users do during the interaction. For example,

- **Eye gaze** can reflect the engagement of the user in a given interaction: users that frequently make eye contact with the agent, or make prolonged eye contact often report a higher engagement [3, 115, 68]. Moreover, since eye contact is also a sign of social connection, some studies consider mutual gaze episodes (when both the agent and the human look at each other) as a metric of rapport [158], or as a factor influencing the decision-making of the human user [10].
- **Task fluency** refers to how coordinated and seamless the joint actions of the human and the agent are in the context of a collaborative tasks [56, 55]. A fluent interaction with high concurrent activity (indicating teamwork) and low idle times often correlates with higher acceptance [5] and trust [101] in the agent.
- **Interaction duration** can also be considered an important metric to assess the quality of an interaction, especially in long-term scenarios. The frequency of voluntary interactions with the agent can be considered an indicator of engagement and acceptance [62, 66].

Several works have built upon these metrics to develop models that predict the user's behavior during the interaction. Guo and Yang [50] and Chen et al. [23]) explore how to build data-driven models of the user's trust for specific collaborative decision-making tasks. Similarly, Lee et al. [82] introduces a computational framework to model the engagement of the human user in a robotic storytelling task. We highlight that agents often do not have access to the internal state of the human and measuring precisely how a user may change as a result of an interaction can be intractable [48]. Yet, predictive methods of human interaction metrics could, in principle, still be used as a proxy measurement. However, we carefully point out two limitations of these approaches: (i) they are still heavily task-dependent, limiting their usefulness for long-term, open interactions such as the ones enabled by coexistence; (ii) building static models of the behavior of the user, as currently done, will, once again, lead to the development of generic and stagnant agents, as detailed in Section 2.

Rather, we join Kamino et al. [61] (see Appendix D, Social robots, for a discussion of this work) to encourage the AI and interaction communities to shift their emphasis from measuring coexistence to *constructing* coexistence. That is, defining the purpose of the artificial agent and refining its performance is meaningful and reciprocal in and of itself. In other words it is not the task-dependent metrics that ensure coexistence but the construction of them.

# D   Are Current Embodied Agents Already Coexisting?

Naturally, one might question whether current embodied agents are already coexisting with humans. In this section, we present examples and discussions on key challenges inhibiting current agents from being coexisting.

**Social robots**   A prominent example of embodied agents designed for human interaction are social robots [17, 85]. Companies like Jibo and Anki introduced social robots to the market with high expectations, only to face eventual failure [138]. A significant factor contributing to this is the challenge of sustaining long-term interactions by current embodied agents. Without the ability to change through interaction and become situated into their environment, social robots remain ill-suited for prolonged use. They often succumb to the *novelty effect*, where user engagement diminishes over

time as the robot's initial appeal wears off [118]. The ethnographic field study on social robots by Kamino et al. [61] highlights a notable exception to this trend. Their empirical evidence suggest that meaningful, long-term interactions with embodied artificial agents has less to do with specific design features. Instead meaningful interactions arise from "interconnected moments of situated interaction, related emotional responses, and meaning-making among people as they interact with robots and each other at different levels of organization." We see this exception as opportunity to design coexisting agents that respond to the broader social and environmental context that they are situated in.

**Bias-amplifying interaction** Large language models have been widely integrated into the architecture of embodied agents [148, 34].These models have now been broadly adopted by diverse user groups. While most AI systems influence human behavior, they themselves do not retain user-driven modifications beyond the immediate context window. This lack of adaptability is already problematic, as user-provided knowledge is not incorporated. Worse, studies have shown that interacting with slightly biased AI systems can amplify biases in users, an effect not observed in human-human interactions [47]. These systems not only fail to adapt through interaction, reinforcing a unilateral dynamic, but they also degrade overall system quality by increasing bias in users. As LLMs are increasingly integrated into interactive robots, these issues are likely to persist, if not worsen, through prolonged human-robot interactions.

**Please, just turn on the light** In industrial settings, robotic failures require expert technicians to debug classifiers, diagnose issues, and retrain models with additional data, such as images captured under varied lighting conditions. Consider now, instead, a robot designed to tidy up homes and offices by identifying, classifying, and sorting objects. Relying on expert interventions, similar to those in industrial settings, is impractical for home-deployed robots. A more viable solution is for robots to make use of humans' situated knowledge within their environment. Humans understand their space and might recognize how the specific lighting affects object classification. Instead of requiring an expert to retrain the system, a robot could ask for help [64], prompting users to turn on the light and even learning that doing so improves classification performance. By adapting through situated interactions, the robot avoids repeated failures and reduces the need for costly expert intervention and large-scale data collection. This behavior realistically adjusts the human's expectations on the agent's capabilities and invites them to accommodate their limitations. Furthermore, we are argue that such situated interactions not only improve performance but drive continual evolution and optimization.

# E   Limitations of Current Learning Algorithms for Coexistence

In this section we argue that, in their current form, standard learning approaches for building embodied artificial agents, designed to overcome the assumption of a static optimization problem, are still insufficient to foster coexistence.

**Meta-learning** is often proposed as a solution for adaptation to new tasks. A meta-learning agent trains over a distribution of tasks so that it can quickly adapt to unseen task at test time [57]. However, instead of assuming a fixed task (as in an MDP), meta-learning assumes a fixed meta-distribution of tasks, defined by the designer, from which both training and testing tasks are drawn [39]. Consequently, if the agent encounters a fundamentally novel task outside this distribution, it may not be able to adapt, as it was never optimized for tasks beyond the anticipated variations.

**Continual learning** algorithms, instead, relax the fixed training assumption of the previous methods by exposing agents to a sequence of tasks over time. Often, continual learning algorithms focus on mitigating catastrophic forgetting and preserving performance across task sequences [144]. However, similar to meta-learning, most continual learning setups still operate under relatively controlled forms of novelty: they assume a sequence of tasks or data that, while possibly distinct, follows a predictable format (for instance, new classes from the same data distribution [67]). The focus is on incremental change, not the *unknown unknowns* [83] agents may face in the real-world.

**Reinforcement learning** (RL) is traditionally formulated through a Markov decision process (MDP), where the state and action spaces, reward and transition functions are defined apriori by the designer [131]. Additionally, the training and testing environment are often assumed to be the same. The goal of the agent is therefore to maximize the total (discounted) reward accumulated while acting on that particular environment. Other extensions, such as partially-observable MDPs to deal with incomplete state information [60], multi-agent MDPs to deal with multiple agents acting in the same

environment [49], domain randomization to deal with environments with different dynamics [25] or observations [153], or robust MDPs to introduce safety guarantees in the behavior of the agent [90], still leave the burden on the researcher to guess the possible perturbations the agent might face in advance. This is often done by formalizing uncertainty in *quantitative terms*: assuming any environment change can be modeled as some stochastic variation in rewards, transition probabilities, or observation noise. However, these approaches are still unable to deal with structural novelty that might lie outside the prior distribution of possible phenomena defined by the designer. Recently, continual reinforcement learning methods, while still severely underexplored [113], have shown some promise in allowing agents to carry out search processes indefinitely [1], producing behavior in response to all past experience [16]. We believe that these approaches could also be used in the future, complementing evolutionary techniques, to allow agents to explore convergent and divergent behaviors.

For an extended argument on the limitations of these methods for coexistence, we refer the reader to Lehman et al. [83].

## F    Potential Coexisting Technology Today

In this section, we highlight several examples of technology with properties that foster coexistence.

**Mutable morphology and locomotion**    Figure 4 shows how the morphology of an agent can be changed to recover from damage and to re-learn how to walk [71]. The agent learns how to walk through periodically inflating and deflating its individual cells, exploiting its own physical shape. Although this does not involve a human user, it demonstrates the value of mutable morphologies. For example, we see great potential in mutable morphology to express various mannerism through different gaits, especially in the context of Vergunst and Ingold's work on the contextual nature of walking [142]. Thus culminating in rich, heterogeneous populations of artificial agents at scale. Figure 5 shows Yamaha's "Motorid", a shape changing, self-balancing motorcycle [52]. It has a twisting chassis and autonomous driving abilities that influence how riding the motorbike feels in real time. This dramatically changes motorcycling from its culture to its engineering principles. Whilst not a child of the RtD method, but rather a concept bike, it balances divergent and convergent themes. Blurring the definition of what is a bike and an autonomous agent.

**Mutable perceptive fields in human-drone interaction**    Figure 6 shows a system that allows humans to shape how a drone senses its environment. So whilst the human understood how to shape the perceptive field of the drone, the behavioral outcomes of their actions were not. This enabled rich explorations with the drone which lead to diverse and meaningful interactions [78].

**Mutable interaction interface**    Figure 7 shows Bewley and Boer's [11] "Blo-Nut", a silicone doughnut that affords the user a blank slate to interact with. The object inflates and deflates and can be programmed to music. Its non-humanoid shape enables interactions to the human in ambiguous ways, which Sandry [122] argues, is an opportunity to build effective communication between humans and artificial agents.

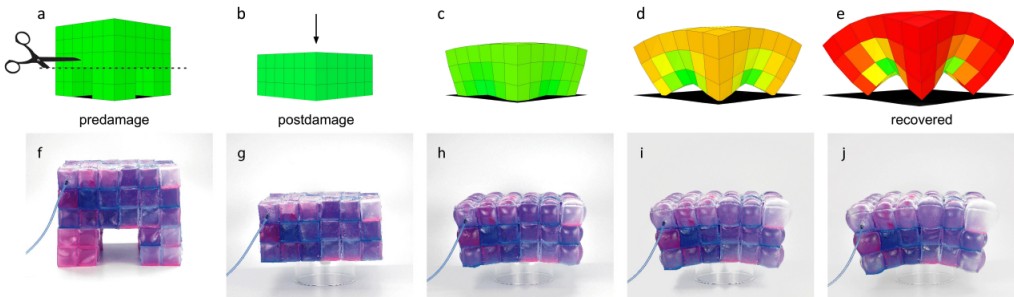

Figure 4: Self recovering locomoting voxels [71]: by virtue of an evolutionary algorithm, the agent is relearning how to walk by changing the inflation patterns of its individual cells. Each change to the physical body is likened to a divergent search for a new and unique locomotion gait. Whilst each improvement in performance in this new body is a convergent search to optimize. Therefore, the agent is able to explore and exploit to it physical environment by simply existing in it.

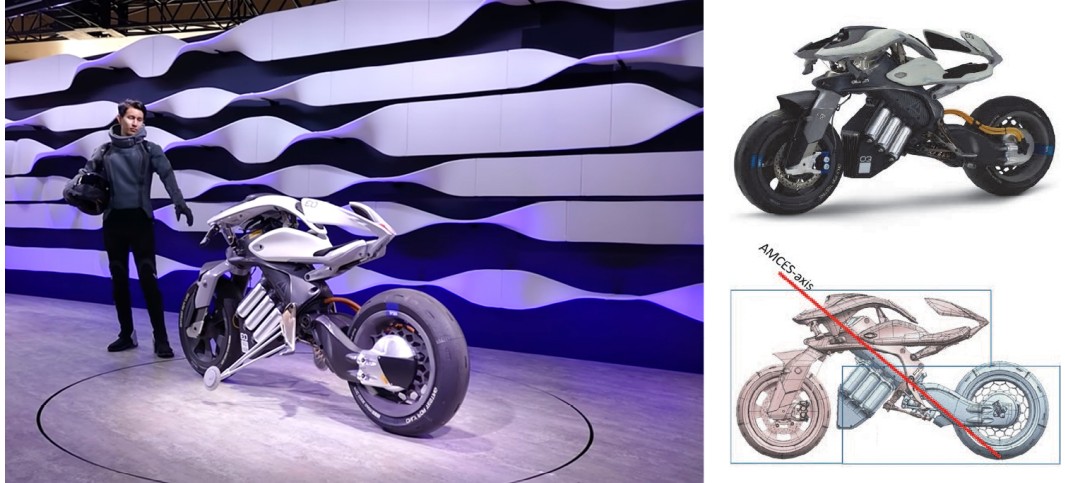

Figure 5: Yamaha's "MOTOROiD" is a shape changing, self-balancing motorcycle [52]. Its unique twisting chassis is able to affect the ride feel in real time as well as drive autonomously.

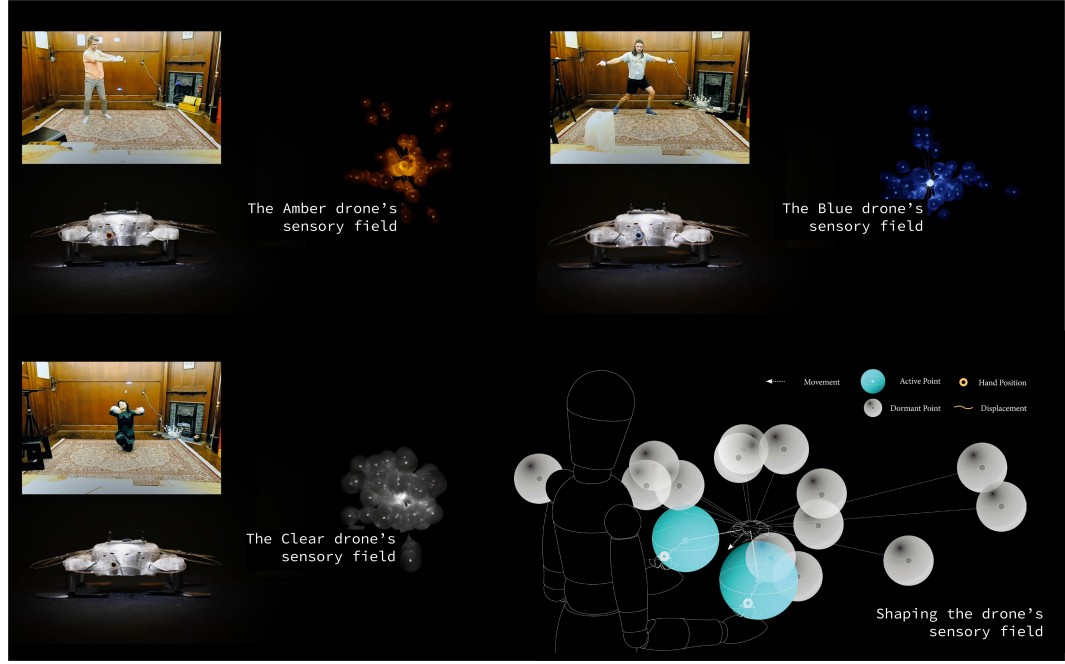

Figure 6: "How to Train Your Drone" [78]: depicted here in orange, clear and blue are the sensory fields of the drones. By interacting with the drone, its sensory field can be changed with human intention. However the consequences of such changes are not always predictable. This work demonstrates the potential of interacting with the sensing and acting capabilities of mutable agents.

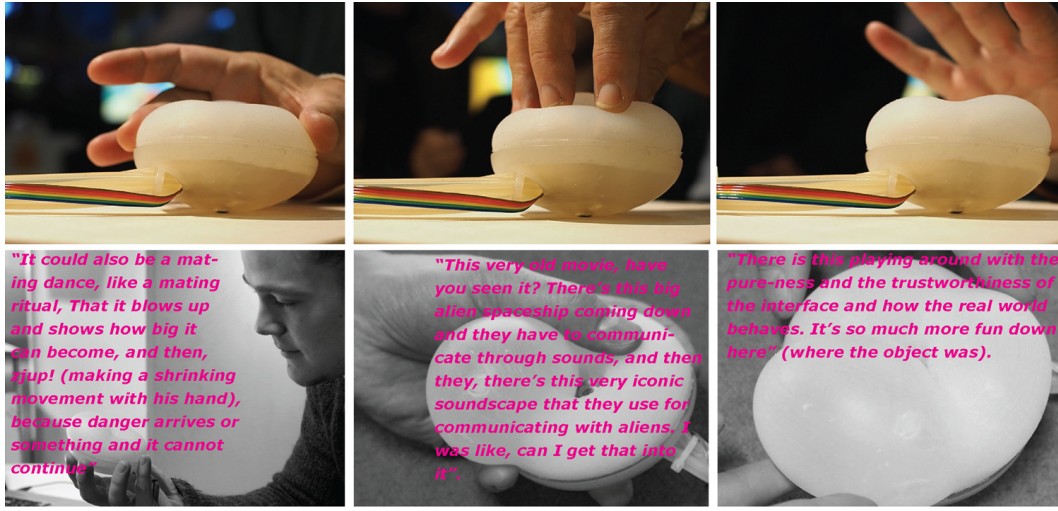

Figure 7: "Blo-Nut" is a silicone doughnut that affords the user a blank slate to interact with [11]. The object inflates and deflates and can be programmed to music.

