# OpenReview forum: "Humans Coexist, So Must Embodied Artificial Agents"
_NeurIPS.cc/2025/Position_Paper_Track — Submitted to NeurIPS 2025 Position Paper Track_

### Official Review · Reviewer_6KG3 · 2025-08-07

**Significance:** 3
**Presentation:** 3
**Rating:** 7
**Confidence:** 3

**Summary:**

The authors argue that the current approach to agent design is unsuitable for long-term, in-the-wild
interaction with humans, and introduce the the need for a new paradigm for coexisting embodied agents. The authors also take an interdisciplinary perspective to understand how human and non-human organisms foster entities that coexist within their specific environments, and provide six key research directions for the AI community to develop coexisting agents. It is a good view for exist and coexist for artificial agents.

**Strengths:**

This topic is very interesting and looking forward. It shows how AI/embodied agents and humans will coexist in the future.

The introduction of "coexistence" as a design paradigm is a fresh contribution, shifting the focus from task optimization to sustained, meaningful interactions. The interdisciplinary analogy to biological systems and design processes is compelling.

This paper highlights the six key research directions toward fostering coexistence and developing coexisting agents, like foster coexistence by embracing open-endedness, embracing the user as the designer, foster coexistence in the space around the agent, foster coexistence within the morphology of the agent, using foundation models as external components, and learning and evolving with humans, these address diverse aspects of coexistence.

**Weaknesses:**

1. Figure 1 does not directly express the main contribution of the paper, and some of the pictures in it are difficult to understand.

2. The references are listed in a random order. Would it be better to number them sequentially?

3. While Appendix C discusses challenges in measuring the quality function, the paper lacks concrete, actionable metrics or methods. This limits the practical testability of the coexistence concept.

**Questions:**

The paper emphasizes "meaningful and reciprocal interactions" as central to coexistence (Abstract and Section 3). Could the authors provide specific examples of what constitutes a "meaningful" interaction in real-world scenarios?

**Alternative Position:**

Yes, and alternative positions are trivial straw-man arguments

**Author Identification:**

No.

**Context:**

3

**Discussion:**

4

**Ethics:**

["NO or VERY MINOR ethics concerns only"]

**Position:**

Yes, the paper argues for or against a position related to machine learning.

**Support:**

3

**Thoroughness:**

3

---

### Official Review · Reviewer_5akx · 2025-08-12

**Significance:** 1
**Presentation:** 3
**Rating:** 5
**Confidence:** 4

**Summary:**

This position paper argues that embodied agents perform well in controlled settings but fail in real-world contexts due to a design paradigm that cannot sustain situated, reciprocal, and long-term interactions with humans and the surrounding environment. The authors call for a paradigm shift toward mutable agents that learn from and influence both users and their environments. To coexist with humans, as humans do with one another by continuously adapting, agents must form reciprocal relationships that leverage and exploit situated knowledge. The behavior should increase the quality of the system. Current systems remain stagnant and generic, relying on static training data or fine-tuned datasets with fixed distributions, that limit adaptation to change. They use generic rather than adaptable general knowledge suited to nuanced contexts. Drawing on analogies to biological organisms, the authors advocate for divergence and convergence in agent design and promote research-through-design as a reflexive, continuous method for enabling reciprocal engagement. They end with a few ethical considerations and limitations to consider, such as malicious actors, and expand the appendix to include examples of coexistence as they define it.

**Strengths:**

The authors present a compelling critique of the current "stagnant" and "generic" approach to designing agents, clearly showing why it cannot support sustainable, long-term interactions with humans. They use this criticism effectively to strengthen their case for a fundamentally different design paradigm for embodied agents, and the argument’s timely relevance extends beyond this argument into broader machine learning, AI, and HCI contexts. Their integration of concepts from evolutionary biology and design theory demonstrates an acknowledgement that agent design requires interdisciplinary engagement. Their introduction of the concept of "steamrolling" is particularly strong, with a clear definition supported by a concrete and memorable example drawn from the prevalence of LLM-style abstracts in computer science papers.

**Weaknesses:**

Below I outline some points that can be improved to strengthen the authors' argument. I include section numbers where necessary.

- **Section 4.1**: The evolutionary biology analogy seems underdeveloped and does not justify why survival-driven adaptation is necessary for embodied agents whose goal is not survival.
- **Section 5.1**: This section is arguably one of the most important but the warning about harmful behavior and the need for ethical and legal frameworks is brief and lacks depth.
- **Appendix C**: I’m concerned that measurements presented for coexistence such as eye gaze and interaction duration might favor neurotypical social interactions and lead to unwanted behavior.
- More examples of meaningful interactions would be helpful.
- The argument does not fully address whether coexistence is necessary or desirable for all types of embodied agents, or why the absence of coexistence would be undesirable (for society).
- The authors missed an opportunity to engage with counterpoints for the existence of coexistence from a critical design perspective and counter-claims that biased humans have and can influence humans and how that would affect coexisting embodied agents.

**Questions:**

Some of the questions below extend from the weaknesses above and would help clarify missing or confusing points:

- Is the underlying assumption that embodied agents must have a pseudo-human role for coexistence to succeed?
- Should coexistence require interpreting social cues and influencing culture, and for which types of agents?
- How does a coexisting embodied agent that can influence humans and the environment improve a system; who defines the boundaries of this influence; and how is this different from steamrolling?
- How would coexisting agents combat "steamrolling" if they are also meant to influence their environment?
- If agents can replace humans at a larger scale, how would coexistence help maintain human stability?
- Why is coexistence desirable and in what contexts would it be undesirable?
- Could mutability lead to harmful outcomes, and when might static data offer better diagnostic stability?
- What happens if coexisting agents learn undesirable behaviors from humans, and who decides what is undesirable?
- How should credit be assigned in coexistence to support continuous evaluation?
- What are the infrastructure costs of developing and sustaining coexisting embodied agents?

**Alternative Position:**

Yes, and alternative positions are trivial straw-man arguments

**Author Identification:**

No.

**Context:**

2

**Discussion:**

3

**Ethics:**

["NO or VERY MINOR ethics concerns only"]

**Position:**

Yes, the paper argues for or against a position related to machine learning.

**Support:**

2

**Thoroughness:**

5

---

### Official Review · Reviewer_txyS · 2025-08-12

**Significance:** 3
**Presentation:** 4
**Rating:** 8
**Confidence:** 3

**Summary:**

This paper advocates for the need for "coexistence" as a necessary paradigm for the development of embodied artificial agents to facilitate meaningful, long-term, in-the-wild interactions with humans that avoid "steamrolling." The authors put forth that the current static state of agents are unsatisfactory since they only operate on predefined tasks and are not able to be adaptive to new stimuli. The paper provides examples across human design and biology of how coexistence is a naturally occurring phenomenon. The alternative to coexistence offered by the paper is "steamrolling," which would lead to homogeneous actions due to agents that do not adapt.  The paper's authors envision a future where embodied agents take a role in shaping and responding to an environment through coexistence.

**Strengths:**

The strengths of this paper are in proposing a change today that would have long-term and real-world impact on humans using embodied artificial agents. The paper is organized clearly and offers examples from biology and design to support its argument.

**Weaknesses:**

This paper does not offer specifics on what coexistence would look like and what that future requires of current practitioners. It makes assumptions about how humans would like to engage with embodied agents without questioning whether that is a future we would even want. The alternatives offered are of homogeneous agents and steamrolling, but do not consider a future where agents are not as fully integrated into human life as the paper assumes. The examples that it provides of coexistence do not feel satisfactory enough to fully support the argument, although the argument they make by itself is compelling. It also does not flesh out the dangers of promoting coexistence for embodied agents, and this is the greatest weakness of the paper.

**Questions:**

Are there any other examples that would support this call for coexistence? What are the dangers of coexistence?

**Alternative Position:**

Yes, and alternative positions are well-considered and addressed by the argument

**Author Identification:**

No.

**Context:**

3

**Discussion:**

4

**Ethics:**

["NO or VERY MINOR ethics concerns only"]

**Position:**

Yes, the paper argues for or against a position related to machine learning.

**Support:**

2

**Thoroughness:**

4

---

### Note · Authors · 2025-09-04

**1-10 Additional Comments:**

We would definitely appreciate a dedicated rebuttal phase for two main reasons: 1) it provides an opportunity to engage deeply with the reviewers and clarify some points they have missed (and that do not correspond to a violation of reviewer conduct); 2) It allows the authors to demonstrate to the reviewers how the discussion brought up by their comments is going to be integrated into the camera ready version of the paper (which is also rewarding for the majority of the reviewers).

**1-11 Submit Again:**

Definitely yes

**1-1 Submission Process:**

4

**1-2 Next Year:**

1) A more structured and predictable schedule for the different stages of the submission period.
2) A dedicated rebuttal phase (before the meta-review period) so that the authors have a chance to discuss with the reviewers.

**1-3 Future Development:**

One potential idea could be to shorten the review period, anticipate the rebuttal phase (if it exists) and, even before the decision, open the forum for discussion with authors of previous NeurIPS position papers, while maintaining the double blind nature of the process.

**1-4 Interest:**

["Panel discussions with other position paper authors", "Structured debates on controversial topics", "Workshops for developing position papers", "Mentorship programs for early-career researchers", "Other (please specify in the next question)"]

**1-4 Other Interest:**

Organizing workshops dedicated to debating and coming out with solutions for problems raised by common position papers.

**1-5 Thoughtful:**

8

**1-6 Supportive:**

8

**1-7 Technical Aspects Versus Position:**

7

**1-8 Gate Keeping:**

8

**1-9 Camera Ready Changes:**

We will address the following main points:

- All Reviewers asked for additional examples of how coexistence might manifest in the future, beyond the existing examples in Appendix F. We will extend this appendix with four forward-looking case-studies that give an intuitive understanding of coexistence, based on recent works in HRI: delivery robots in hospital settings (inspired by Kamino et al., 2024), robot tutors in school settings (Chandra et al., 2025), human-robot art creation (Lee and Ju, 2024), and robot companionship in domestic environments (Kamino et al., 2024). These scenarios envision the novel technologies involved, interactions that emerge, as well as their societal and ethical ramifications.

- At the request of Reviewers txyS & 5akx we will clarify and expand Section 5.1 with further potential risks associated with coexistence. We will focus on the liability in case of misuse of coexisting technology (Bommasani et al., 2025), and the balance between adaptation to local systems and regulatory mechanisms at a higher level (e.g., laws and customs of a country). In addition, we will discuss how to design coexisting systems while avoiding and mitigating potential risks of their usage (e.g., disinformation, cybersecurity, violent behavior).

- We have considered both AGI (as a long-term goal) and value alignment (as a methodology) as alternative positions to coexistence. As Reviewers 6KG3 & 5akx did not appreciate the depth of our alternative positions, we will clarify and expand Section 6. Regarding long-term goals, we will discuss the AGI alignment dilemma (Hellrigel-Holderbaum and Dung, 2025), expanding on the benefits and the risks of aligned AGI systems. For methodology we will discuss recent work in pluralistic alignment (Sorensen et al., 2024; Zhang et al., 2025), and the connections to Arrow’s impossibility theorem (Arrow, 1951).

**3-1 Review Response1:**

txyS

**3-2 Reaction To Review1:**

The reviewer has engaged deeply with our work, and understood the main contributions and arguments of our position. In addition the reviewer also supports our position. Despite this support, the reviewer suggested to include in the manuscript additional concrete real-world examples of coexisting systems and an extended discussion on the ethical considerations of coexistence, which we address in the changes for the camera ready. Additionally, we did not feel any level of gatekeeping from the reviewer, which is welcomed since our paper is highly multidisciplinary in nature. This interdisciplinary nature of our position, we think, pertains to the low confidence scores seen in our reviews and the reviewer’s need for more concrete examples.

**3-3 Review Response2:**

5akx

**3-4 Reaction To Review2:**

We believe that the reviewer has engaged positively (without gate keeping) and shown strong support of our position and work (as highlighted in the Strengths section of the review). However, we believe that, still, the reviewer has not fully grasped some points of our position. For example, the reviewer questions whether the absence of coexistence would be undesirable (W5), which we thoroughly motivate in Section 2 with the societal consequences of non-coexisting systems. The author also questions the link between survival in evolutionary biology and coexisting agents (W1): in Section 4.1., we describe how biology is shaped by interactions with the physical world beyond just survival. In other words, not all biological mechanisms are subject to selection pressure, but they all contribute to the varied natural ecosystems we experience today. This is in stark contrast to the homogenous agents currently steamrolling our everyday life. Additionally, the “missed opportunity” (W6) regarding critical design fails to understand that coexistence invites all stakeholders to engage critically with the technology they interact with. Divergent exploration of an agent (be it behaviourally, morphologically or environmentally) prompts reflection on norms and sense making, this is inherently critical. We will clarify this in Section 4.2. We also felt that the reviewer often focused unnecessarily on technical issues such as concrete measurements of coexistence (W3), which we ourselves critique in Appendix C, and infrastructure costs of developing and sustaining coexisting embodied agents (Q10), which, at this point in time where embodied agents are still not widespread, could only be answered speculatively. While fundamental for concrete implementations of coexistence, we believe these details go beyond the scope of our position. Beyond these points, the reviewer gave insightful suggestions to improve and clarify our position, which we address in “Changes for the Camera Ready”.

**3-5 Review Response3:**

6KG3

**3-6 Reaction To Review3:**

The reviewer engaged well with the position and they strongly support it. The reviewer’s critiques are more surface level (they criticise mainly formatting and figures), and therefore are easily remedied. They also request actionable metrics and methods with the aim of practical testability, which shows that they are more focused on technical aspects as opposed to the position itself. The reviewer asks for examples of meaningful interactions in real life, which was also requested by other reviewers, and we will address this in camera ready version.

---

### Meta-Review · Area_Chair_StFf · 2025-09-08

**Rating:** 7
**Confidence:** 4

**Strengths:**

The reviewers generally acknowledge the position argued by the paper compelling and having strong potential for having impact on robotics and AI. They find the paper clear and timely and appreciate the arguments/examples supported from biology and design theory.

**Weaknesses:**

Despite the positive feedback, one reviewer had concerns about the specifics of the proposed paradigm for practitioners. The other reviewers provided comments and feedback about some of the sections and the figures in the paper.

**Questions:**

See the individual reviews.

**Ethics:**

No.

**Thoroughness:**

1

---

### Decision · Program_Chairs · 2025-09-26

Reject